# Effects of Yeast Culture Supplementation on Growth Performance, Nutrient Digestibility, Blood Metabolites, and Immune Response in Geese

**DOI:** 10.3390/ani12101270

**Published:** 2022-05-15

**Authors:** Jie Zhang, Hang He, Yancong Yuan, Kun Wan, Longjiao Li, Anfang Liu

**Affiliations:** 1College of Animal Science and Technology, Southwest University, Chongqing 402460, China; zhangjie813@163.com (J.Z.); y1400510152@163.com (Y.Y.); iforson03@163.com (K.W.); 2College of Animal Science and Technology, Chongqing Three Gorges Vocational College, Chongqing 404155, China; swu_hh@163.com (H.H.); 15123579066@163.com (L.L.)

**Keywords:** yeast culture, growth performance, digestibility, immune function, geese

## Abstract

**Simple Summary:**

Antibiotics benefit animal growth performance and health. However, an overuse of antibiotics can lead to antibiotic residue and the development of drug-resistant bacteria. Considering the possible serious consequences of antibiotic overuse, non-antibiotic alternatives, such as yeast and yeast products, are widely used in animal diets to improve growth and productive performance. Geese are important poultry, and their meat provides high-quality protein to humans, but studies showing how yeast culture affect geese are limited. On this basis, the effects of dietary yeast culture supplementation on growth performance, nutrient digestibility, blood metabolites, and immune functions of geese were studies. The results showed that the dietary addition of yeast culture could improve growth performance and nutrient digestibility and can modulate the immune response of geese. This dietary strategy based on feed additives is an effective method for improving the growth efficiency of the geese.

**Abstract:**

The present study was conducted to investigate the effects of dietary yeast culture (YC) supplementation on growth performance, nutrient digestibility, blood metabolites, and immune functions in geese. One-day-old Sichuan white geese (*n* = 300) were randomly divided into five groups containing 0 (control), 0.5%, 1.0%, 2.0%, and 4.0% of YC in the diet for 70 days. In general, the dietary supplementation of YC significantly increased the average daily gain and feed conversion ratio (*p* < 0.05) in which the 1.0% or 2.0% levels were better and significantly reduced the average daily feed intake at the 2.0% level (*p* < 0.05). YC supplementation increased digestibility of P (quadratic, *p* = 0.01) and gross energy (quadratic, *p* = 0.04) from days 23 to 27 and crude protein from days 23 to 27 and days 64 to 68 (quadratic, *p* ≤ 0.05), with the 2.0% level being the most effective. Serum metabolites were significantly affected by dietary YC (*p* < 0.05). Supplemental YC increased IL-2 on day 28 (linear, *p* = 0.01; quadratic, *p* = 0.04) and lysozyme on day 70 (quadratic, *p* = 0.04) and decreased complement C4 on day 70 (linear, *p* = 0.05). Interferon-γ, interleukin-2, and tumor necrosis factor-α genes were mostly up-regulated after YC supplementation, and interferon-γ and interleukin-2 gene expression levels were significantly increased at the 2.0% level (*p* < 0.05). Taken together, dietary YC supplementation improved growth performance and affected nutrient digestibility, serum metabolites, and immune function in geese, which was optimized at the 2% YC level in the present study.

## 1. Introduction

With the rapid development of large-scale and intensive animal husbandry, animal diseases have become more complex and diverse, and veterinary drugs and antibiotic feed additives are widely and increasingly being used. However, frequent and excessive antibiotic use is an increasing concern because of possible antibiotic residues, the development of drug-resistant bacteria, and the inhibition of the animal’s immune system [1,2,3]. Therefore, novel dietary supplements, such as probiotics, prebiotics, and other feed additives, have been used to replace antibiotics to improve animal health and maintain production efficiency [4,5,6]. Yeast cultures (YCs) are widely used as excellent feed additive candidates in animals [7]. YCs are microecological products formed by yeast after sufficient anaerobic fermentation on specific media and are mainly composed of yeast metabolites, fermented medium, and a few yeast cells [3,8]. YCs are rich in vitamins, saccharides, minerals, enzymes, growth-promoting factors, and amino acids, which may benefit animal growth, metabolism, and health [8,9].

Many studies have reported the application of YCs in animals because of the potential benefits to animal production yields. For example, supplementing dairy cow diets with YCs improved lactation performance [10,11,12] and increased feed efficiency [13]. Supplemental YCs increased the carcass weights of calf-fed Holstein steers throughout the growing-finishing period [14]. Supplementing buffaloes with YC increased milk production by increasing the feed, energy, and nitrogen conversion efficiency [15]. Adding YCs significantly affected the ruminal pH in rams [16] and helped prevent and treat acute ruminal lactic acidosis in sheep [17]. In horses, dietary YC supplementation increased variability in Streptococcus bovis [18] and aided in fiber digestion [19]. In pigs, the dietary supplementation of YC improved growth performance and may be a good alternative to antibiotics [3,8]. YC feeding benefitted the gut microbiota, growth, and biochemical parameters in grass carp [7] and promoted feed intake, weight gain, and disease prevention in Nile tilapia [20]. YC supplementation improved growth performance and affected immune functions, Ca and P digestibility, and intestinal mucosal morphology in broilers [4] and increased egg weight and decreased egg yolk cholesterol without affecting performance or egg traits [9].

In summary, YCs have been extensively studied in livestock, especially ruminants, and several studies have been conducted on aquatic animals and poultry. However, no study has reported dietary YC supplementation in geese. Geese are economically important herbivorous waterfowl in central Europe and Asia, especially in China (90% of the world’s production), and are characterized by rapid growth, high fecundity, and strong disease resistance [21]. Geese provide nutritious meat and eggs worldwide, are a valuable protein source for humans, and provide high-quality liver fat and feathers [22]. Because of its many benefits, we hypothesized that YC supplementation would influence performance in geese by improving digestion and immune functions. Therefore, this study was conducted to evaluate the effects of YC dietary supplementation on growth performance, nutrient digestibility, blood biochemical parameters, and immune functions in geese.

## 2. Materials and Methods

### 2.1. Animals, Diets, and Experimental Design

In total, three hundred healthy Sichuan white goose (mixed sex) at 1-day-old age having an average body weight (BW) of 95.57 ± 2.42 g were used and randomly divided into five groups containing 0 (control), 0.5%, 1.0%, 2.0%, and 4.0% of a commercial YC (Baihuibang, Beijing Enhalor Biotechnology Co., Ltd., Beijing, China) product in the diet. YC is formed by *Saccharomyces cerevisiae* (Sa-10) through a two-step continuous fermentation of liquid and solid. In the process of aerobic liquid fermentation, *Saccharomyces cerevisiae* cells fully proliferate. Then, the fermentation broth is injected into a medium composed of a variety of high-quality grains for anaerobic solid fermentation. The ingredient analysis shows that YC contains ~15.0% CP, ~3.5% crude fat, ~8.7% crude fiber, ~14.2% amino acid, ~3.3% mannan, ~14.0% β-glucan and rich in deformed medium, yeast cell walls, cell contents, and vitamins. The basal diets with corn-soybean meal in recommendations of the NRC (1994) [23] for starter (days 1–28) and for grower (days 29–70) periods were formulated (Table 1). Geese were housed from April to June in pens (3.5 m × 3.0 m) and reared in a windowed poultry house. Each group consisted of 6 replicates with 10 geese per pen. Geese were allowed access to feed (in pellet form) and water *ad libitum* throughout the entire experimental period. Feed was provided four times daily at 7:30, 12:30, 17:00, and 21:00 h. The house average temperature during the experiment period was (21.32 ± 2.39) °C, and relative humidity was (84.03 ± 5.15)%. All geese management was consistent with recommendations of goose feeding management technical procedures.

### 2.2. Sampling and Analyses

#### 2.2.1. Growth Performance

Every morning the feed consumption before the feeding was recorded and BW was measured on day 1, 28 and 70 with empty stomach. Then calculated the average daily gain (ADG), average daily feed intake (ADFI), and feed conversion ratio (FCR) for starter, grower and overall periods. Mortality was recorded when occurred during the experiment, including mortality date, BW, mortality reason, and feed consumed.

#### 2.2.2. Nutrient Digestibility

The determination of nutrient digestibility was conducted by collecting total fecal once daily for 5 consecutive days from day 23 and again from day 64, as previously described [4] with minor modifications. In brief, total excreta collection was made from 3 geese in metabolic cages that were randomly selected in each pen. Metabolic cages were used only during excreta collection periods, and they were removed afterward. Excreta from each pen were collected, mixed, and weighed, and a 10% aliquot was sampled and frozen in −20 °C. Daily aliquots of excreta were combined per pen within each period, and a 10% aliquot was taken and dried in 70 °C. Dried excreta samples were crushed by a pulverizer and passed through a 40-mesh sieve and stored in a glass vial. Excreta samples were analyzed for Ca, P, crude protein (CP), and gross energy (GE).

#### 2.2.3. Serum Metabolites

Blood samples of 18 fasted geese randomly selected from each dietary treatment (3 geese per pen) were collected from vena brachialis under the wing at day 28 and 70 for a biochemical parameter assay. Serum samples were collected by centrifugation at 3500× *g* for 10 min and stored at −80 °C. Serum samples were analyzed for aspartate aminotransferase (AST), alanine aminotransferase (ALT), alkaline phosphatase (ALP), total cholesterol (TC), triglycerides (TG), high density lipoprotein (HDL), and low-density lipoprotein (LDL) using a CL-8000 clinical chemical analyzer (Shimadzu, Kyoto, Japan) via standard enzymatic procedures.

#### 2.2.4. Immune Factors

Serum lysozyme activity was measured by the method of turbidimetry using micrococcus lysodeikticus cells as substrate. Serum immunoglobulin G (IgG), immunoglobulin A (IgA), immunoglobulin M (IgM), complement C3, and complement C4 were measured by immunoturbidimetry using monoclonal antibodies. Serum interleukin-2 (IL-2) level was determined by antigen competition method of ELISA. All above tests were carried out using commercial kits (Solarbio, Beijing, China).

#### 2.2.5. Immune Genes

On day 70, 18 geese from each treatment (3 geese per pen) were killed by cervical dislocation for measurement of spleen interferon-γ (*IFN-γ*), interleukin-2 (*IL-2*), and tumor necrosis factor-α (*TNF-α*) genes expression using quantitative PCR (q-PCR). Total RNA was extracted from spleen using TRIzol reagent (Invitrogen, Carlsbad, CA, USA) and further purified using an RNeasy column (Qiagen) according to the manufacturer’s protocol. cDNA was synthesized using the oligo (dT) 6-mer primers provided in the PrimeScript RT Master Mix kit (TaKaRa, Dalian, China). q-PCR was performed using the SYBR Premix Ex Taq kit (TaKaRa) on a CFX96 Real-Time PCR detection system (Bio-Rad). The house-keeping gene (β-actin) used as the endogenous control and target gene primer sequences used for q-PCR are shown in Table 2. All measurements were performed in triplicate, and negative controls (no cDNA template) were always included. The relative amounts of objective genes were calculated using the ΔΔCt method.

### 2.3. Statistical Analysis

All data were analyzed using one-way ANOVA of SPSS 22.0 software package for Windows (IBM Corporation, 2014) with LSD multiple comparison tests. The effect of supplemental levels of YC was determined using orthogonal polynomials for linear and quadratic effects. Variability in data is expressed as the SE of means, and data were expressed to be statistically significant when a probability level of *p* ≤ 0.05 was attained.

## 3. Results

### 3.1. Growth Performance

As shown in Table 3, dietary YC supplementation significantly affected the performance of BW, ADG, ADFI, and FCR of geese (*p* < 0.01), with the exception of ADFI during the starter period (days 1–28, *p* > 0.05). Compared with the control group, the YC-treated group showed improved ADG and FCR, whereas at the 2.0% level, YC significantly reduced ADFI during the grower (days 29–70) and overall (days 1–70) periods (*p* < 0.05). No differences in ADG and FCR were found between the YC-treated group at 1.0% and 2.0% (*p* > 0.05), specifically not at the 0.5% and 4.0% levels, which were more effective (*p* < 0.05) during the starter period. ADG was greater and ADFI and FCR were lower at the 2.0% level than at the 0.5%, 1.0%, and 4.0% levels during the grower and overall periods (*p* < 0.05), whereas the 0.5%, 1.0%, and 4.0% levels did not differ (*p* > 0.05). As the dietary YC level increased, BW, ADG, and FCR tended to be improved during the starter period (quadratic, *p* = 0.06), with the 1.0% and 2.0% levels being more effective. Mortality during the experimental period was 1.0% (*n* = 3) and was unaffected by the dietary treatments (*p* > 0.05).

### 3.2. Nutrient Digestibility

The digestibilities of P and GE from days 23 to 27 and Ca and CP from days 64 to 68 were significantly affected (*p* < 0.05) by dietary YC supplementation (Table 4). Compared with the control group, Ca digestibility from days 64 to 68 and GE digestibility from days 23 to 27 were significantly decreased (*p* < 0.05) at the 4.0% level; at the 1.0% and 2.0% levels, YC significantly increased (*p* < 0.05) P and GE digestibility from days 23 to 27 and CP digestibility from days 64 to 68. With the increase in dietary YC, P and GE digestibility from days 23 to 27 and CP digestibility from days 23 to 27 and days 64 to 68 exhibited quadratic responses (*p* ≤ 0.05), with the 2.0% level being the most effective.

### 3.3. Serum Metabolites

YC dietary treatments significantly affected (*p* < 0.01) the serum metabolite parameters on days 28 and 70 (Table 5). The concentration of ALT, ALP, and TG in the YC-treated groups significantly increased (*p* < 0.05) compared with those of the control group, whereas TC and HDL significantly decreased (*p* < 0.05) in general. Interestingly, the YC-treated groups had lower AST and LDL concentrations than did the control group on day 28 but greater AST concentration on day 70. Furthermore, the inclusion of YC tended to increase ALT (linear, *p* = 0.05), AST (linear, *p* = 0.07), and TG (linear, *p* = 0.06) concentrations on day 70.

### 3.4. Immune Factors

Adding YC to the geese diets significantly influenced (*p* < 0.05) the response of lysozyme, IL-2, IgM, complement C3, and complement C4 on days 28 and 70 (Table 6); however, the level of IgA was unaffected (*p* > 0.05). Dietary YC significantly affected IgG on day 70 (*p* < 0.01), but this response was not observed on day 28 (*p* > 0.05). Compared with the control group, YC supplementation significantly increased IL-2 and IgM at the 0.5% level; the lysozyme, IL-2, and IgM at the 2.0% level; and IgM at the 4.0% level on day 28 and on day 70 (*p* < 0.05). Moreover, 1.0% level also significantly increased IL-2 on day 28 and the lysozyme and complement C4 on day 70 (*p* < 0.05). However, 1.0% YC level significantly decreased the lysozyme on day 28 and IgG on day 70 (*p* < 0.05), and complement C4 on days 28 and 70 significantly decreased at the 2.0% and 4.0% levels (*p* < 0.05), respectively. As dietary YC increased, IL-2 increased on day 28 (linear, *p* = 0.01; quadratic, *p* = 0.04), and complement C4 decreased on day 70 (linear, *p* = 0.05). Lysozyme exhibited quadratic responses (*p* = 0.04) on days 70, with the 2.0% level being the highest.

### 3.5. Immune Genes

Quantitative evaluation indicated that dietary YC supplementation significantly affected (*p* < 0.05) immune gene expression levels (Figure 1). Compared with the control group, *IFN-γ* gene expression levels significantly increased at 1.0% and 2.0% (*p* < 0.05) and decreased at 4.0% (*p* < 0.05). *IL-2* gene expression levels increased significantly at 0.5%, 1.0%, and 2.0% (*p* <0.05), and *TNF-α* gene expression levels were greater at 0.5%, 2.0%, and 4.0% (*p* < 0.05). *IFN-γ* and *IL-2* gene expression levels were the highest at 2.0%, moderate at 0.5% and 1.0%, and the lowest at 4.0%. The *TNF-α* gene expression level tended to increase linearly (*p* = 0.06) with increasing levels of YC.

## 4. Discussion

In this study, YC supplementation did not affect goose mortality (*p* > 0.05), which is consistent with previous studies on laying hens [9], broilers [4], quails [24], and calf-fed Holstein steers [14]. Conversely, Tangendjaja [25] reported that YC supplementation in laying hens significantly reduced mortality (*p* < 0.05). Similarly, Fathi et al. [26] found that broilers fed a diet supplemented with 1.25 g/kg of YC had lower mortality rates throughout the experiment (*p* < 0.05). Barens et al. [27] indicated that mortality was significantly reduced in rainbow trout during the first 4 weeks of rearing with YC-supplemented diets (*p* < 0.05). The specific mechanisms of YC on health and/or viability remained unclear because the results are affected by multiple factors such as species, environment, and feed.

In the current study, supplemental YC improved body weight gain (BWG) and FCR in geese. In poultry, YC research mainly focused on chickens, and the results suggest that YC benefits BWG and FCR [2,4,9,26]. At appropriate YC levels, BWG was unattributed to increased feed consumption, and even Liu et al. [28] indicated that 0.2% dietary YC supplementation decreased feed intake in laying hens. Studies with livestock on cattle [14,29,30], sheep [31], and pigs [3,8,32] also demonstrated that dietary YC supplementation increased ADG. These benefits may be because YC supplementation influences gastrointestinal microbiota and intestinal morphology to improve nutrient digestion and retention [3,10,33], and it increases the relative abundances of cellulolytic, amylolytic and lactate-utilizing bacteria [34]. Additionally, YCs contain peptides, organic acids, oligosaccharides, amino acids, and other factors that promote animal growth [4]. In the present study, BWG and FCR tended to be best at the 2.0% level, possibly because YC interacted with the gastrointestinal system by triggering proper levels. Similarly, low YC levels would more effectively improve immune response performance because demand for the immune response is minimal under low disease challenge or stress conditions [4]. However, other studies reported that YC supplementation did not increase BWG [15,35,36]. Different responses may be related to different dietary nutritional densities, durations of supplementation, and yeast product formulations, making comparisons among studies difficult [4,15].

In the present study, YC supplementation improved Ca, P, CP, and GE digestibilities, with better responses at the 2.0% level for P, CP, and GE. Several studies have demonstrated that probiotics, including YCs, are one approach to improving nutrient digestibility. Shen et al. [8] indicated that dietary YC supplementation improved GE and CP digestibility in nursing pigs. Gao et al. [4] and Bradley et al. [37] showed that supplemental YC improved the utilization of digested P and Ca in poultry. This improvement in nutrient utilization may have been due to YC constituents, including phytase [4], oligosaccharides [38], and growth factors [39]. Previous studies reported that increased villus/crypt ratios [40] and feed retention times [41] exert positive effects on nutrient digestibility. The effect of YC on the villus/crypt ratio and feed retention times for geese was not measured in our experiment and, thus, warrants further research. However, other studies showed that YC did not affect energy or protein digestibility [4,36]. Dias et al. [13] found that YC supplementation did not affect the total-tract apparent digestibility of nutrients in dairy cows.

Blood metabolite values are frequently used to assess growth and general metabolic health status in animals [15,42]. For example, serum ALT and AST are used to indicate liver damage; TC, TG, HDL, and LDL are used to indicate fat mobilization, and ALP is used to indicate growth [15]. The ALT and AST results in the present study indicated that YC supplementation negatively affected liver health, possibly because the liver had not adapted to the rapid growth, which was confirmed by ALP. Studies on laying hens [9], buffaloes [15], lactating ewes [43], dairy cows [44], goats [45], and rams [16] reported that YC supplementation did not affect serum ALT or AST. Nursoy and Baytok [46] found increased TG in YC-fed dairy cows, but Chen et al. [2] indicated that TG was unaffected in YC-fed broiler chickens. Research on rabbits [47] and broiler chickens [48,49] showed that probiotic supplementation reduced TC and LDL, which was consistent with the results of the present study. However, other studies reported no difference in TC or LDL levels between probiotic-supplemented and control groups [2,50]. Evidence has shown that probiotics and their components regulate host lipid metabolism via active substances that deconjugate bile salts in the intestines, thereby preventing bile salts from acting as precursors in cholesterol synthesis [51,52]. However, other studies reported inconsistent results, likely because of differing doses, ration compositions, animal species, animal ages, and supplementation strains.

Studies have reported that yeast supplementation modulated immune responses in humans [53], chicks [4], pigs [8], dairy cows [54], and fish [20] to maintain health and improve growth performance by reducing pathogenic bacteria and improving gut health. Similarly, cellular studies also support the role of YC in innate immune functions [55]. In this study, dietary YC increased the serum lysozyme, IL2, IgM, and complement C3 contents, particularly at the 0.5% and 2.0% levels, which is consistent with the findings of Gao et al. [4] and Fathi et al. [26] who reported increased lysozyme and IgM in YC-fed birds. Studies have shown that probiotics (e.g., YCs) stimulate animal immunity via the following methods: (i) flora from the probiotics migrate throughout the gut wall and multiply to a limited extent; (ii) dead organisms release antigens that are absorbed and stimulate the immune system [51]; (iii) the luminal and mucosal gut microbiotas are modulated and protected against inflammation [53]; (iv) yeast cell wall polysaccharides can positively affect immune functions [54]; (v) harmful and pathogenic microorganisms are competitively excluded [56]; (vi) probiotics benefit the intestinal environment and improve intestinal morphology [3,57]; (vii) probiotics help maintain a physiological balance of immunopotent cells, thus providing a healthy environment for the immune system; (viii) oligosaccharides in yeast cell walls bind to viruses and function as vaccine adjuvants to increase antibody titers [26]. Surprisingly, YC did not affect IgA and even reduced IgG in the present study. The reason for this is unknown, and further studies are necessary to interpret these results.

Dong and Wang [58] indicated that the dietary prebiotic significantly increased immune gene expressions in red swamp crayfish, suggesting that YC potentially affects immune genes. In the present study, YC supplementation upregulated three immune genes, *IFN-γ*, *IL-2*, and *TNF-α*, in the spleen, especially at the 2% level, with *IFN-γ* and *IL-2* exhibiting the highest expressions. This result is supported by the protein study of Shen et al. [8], who reported that YC increased cytokine IFN-γ production by triggering a Th-1 response in the gut. IFN-γ and IL-2 activate macrophages, while TNF-α is produced by macrophages. Therefore, the results may be explained in that YC improves animal immunity by activating macrophages [4,8] that can rapidly and efficiently phagocytize the bacteria.

Based on the response of the measured indicators to different levels of YC, the results showed that the optimal YC level was 2% on the whole, and 4.0% YC was not more effective in the present study, which may be due to the high levels of YC causing geese to develop immune tolerance, leading to the wastage of energy and nutrients and suppressing growth performance [59].

## 5. Conclusions

Dietary YC supplementation improves growth performance and affects nutrient digestibility, serum metabolites, and immune function of geese. Growth performance was the best at the 2% YC supplementation level under the experimental conditions of this study. These results may be attributed to YC modulating the animals’ immune status.

## Figures and Tables

**Figure 1 animals-12-01270-f001:**
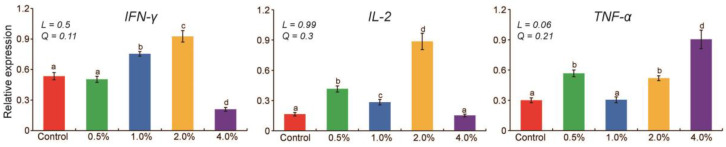
Effect of yeast culture supplementation in goose diets on immune gene expression. Means with different letters differ significantly (*p* < 0.05). Orthogonal contrasts: L = linear and Q = quadratic effect of yeast culture supplementation.

**Table 1 animals-12-01270-t001:** Composition and nutrient content of basal diets (dry basis).

Ingredients (%)	Starter (Days 1–28)	Grower (Days 29–70)
Corn	63.80	53.60
Wheat bran	2.99	14.50
Soybean meal	20.00	11.50
Rapeseed meal	4.00	/
Rice bran	/	13.40
Silkworm chrysalis	4.30	1.79
CaHPO_4_	1.59	0.90
Limestone	0.87	0.75
L-Lysine (98.5%)	0.15	0.18
DL-Methionine	0.05	0.07
Salt (NaCl)	0.20	0.20
Choline chloride	0.05	0.12
Premix ^1^	2.00	2.00
Sand	/	1.00
Total	100	100
Nutrient content		
Metabolizable energy (MJ·kg^−1^) ^2^	11.97	11.21
Crude protein (%)	20.43	14.81
Crude fiber (%)	4.12	8.04
Calcium (%)	0.87	0.80
Available P (%) ^2^	0.43	0.40
Lysine (%)	1.14	0.85
Methionine (%)	0.36	0.30

^1^ The premix provides the following per kilogram of diet: VA 40,000 IU, VD_3_ 2000 IU, VE 60 mg, VK_3_ 2 mg, VB_1_ 4 mg, VB_2_ 24 mg, VB_6_ 4 mg, VB_12_ 50 μg, nicotinic acid 12 mg, pantothenic acid 36 mg, folic acid 4 mg, biotin 0.4 mg, Fe 120 mg, Cu 4 mg, Mn 300 mg, Zn 160 mg, I 0.2 mg, and Se 0.2 mg. ^2^ Calculated values.

**Table 2 animals-12-01270-t002:** Primer sequences used for quantitative PCR.

Gene Name ^1^	Primer Sequences (5′→3′)	Product Size (bp)	GenBank No.
*IFN-γ*	F: ACATCAAAAACCTGTCTGAGCAGC	94	AF087134
	R: AGGTTTGACAGGTCCACGAGG		
*IL-2*	F: AATACTAGCACAGAGACAACCAG	768	AF294323
	R: TTACTGAAATTTATTAAATATCATCTA		
*TNF-α*	F: GAATGAACCCTCCTCCG	139	EU375296
	R: ATCTGGTTACAGGAAGG		
β-actin	F: CAACGAGCGGTTCAGGTGT	92	M26111
	R: TGGAGTTGAAGGTGGTCTCGT		

^1^ *IFN-γ*: interferon-γ; *IL-2*: interleukin-2; *TNF-α*: tumor necrosis factor-α.

**Table 3 animals-12-01270-t003:** Effect of supplemental yeast cultures in geese diets on growth performance.

Item	Yeast Culture Supplementation (% of Diet)					SEM	*p*-Value	
	Control	0.5%	1.0%	2.0%	4.0%		L	Q
Mortality (goose)	0	1	1	0	1	0.26	/	/
BW (g/goose)								
d 1	94.50	96.33	94.42	95.63	96.67	1.60	0.24	0.57
d 28	1001 ^a^	1090 ^b^	1184 ^c^	1172 ^d^	1000 ^a^	2.32	0.78	0.06
d 70	2868 ^a^	3025 ^b^	3027 ^b^	3474 ^c^	3063 ^b^	46.82	0.52	0.21
ADG (g/goose)								
d 1 to 28	32.39 ^a^	35.48 ^b^	38.93 ^c^	38.43 ^c^	32.27 ^a^	0.11	0.77	0.06
d 29 to 70	44.44 ^a^	46.07 ^ab^	45.86 ^a^	54.82 ^c^	49.12 ^b^	1.14	0.33	0.29
d 1 to 70	39.62 ^a^	41.83 ^b^	41.89 ^b^	48.26 ^c^	42.38 ^b^	0.66	0.52	0.21
ADFI (g/goose)								
d 1 to 28	52.10	51.51	51.92	51.72	51.60	0.58	0.39	0.72
d 29 to 70	227.4 ^a^	226.0 ^a^	226.2 ^a^	221.2 ^b^	227.5 ^a^	1.14	0.94	0.27
d 1 to 70	157.3 ^a^	156.2 ^a^	156.5 ^a^	148.0 ^b^	157.2 ^a^	0.87	0.83	0.37
FCR								
d 1 to 28	1.61 ^a^	1.45 ^b^	1.33 ^c^	1.35 ^c^	1.60 ^a^	0.02	0.78	0.06
d 29 to 70	5.12 ^a^	4.92 ^ab^	4.93 ^ab^	3.87 ^c^	4.63 ^b^	0.12	0.39	0.29
d 1 to 70	3.97 ^a^	3.74 ^b^	3.74 ^b^	3.07 ^c^	3.71 ^b^	0.07	0.56	0.22

Note: Means within the same row with different superscript letters differ significantly (*p* < 0.05). Orthogonal contrasts: L = linear and Q = quadratic effect of yeast culture supplementation.

**Table 4 animals-12-01270-t004:** Effect of yeast culture supplementation in goose diets on nutrient digestibility.

Digestibility (%)	Yeast Culture Supplementation (% of Diet)					SEM	*p*-Value	
	Control	0.5%	1.0%	2.0%	4.0%		L	Q
Ca								
d 23 to 27	30.25	30.55	31.06	30.35	29.05	2.59	0.12	0.09
d 64 to 68	18.04 ^a^	16.62 ^ab^	18.18 ^a^	17.26 ^a^	15.32 ^b^	0.54	0.11	0.29
P								
d 23 to 27	44.21 ^a^	48.04 ^ab^	50.02 ^bc^	53.35 ^c^	45.34 ^a^	1.31	0.96	0.01
d 64 to 68	50.63	49.36	51.46	54.73	53.38	1.11	0.17	0.29
CP								
d 23 to 27	67.47	71.53	73.41	74.64	72.67	2.04	0.34	0.05
d 64 to 68	65.04 ^a^	70.26 ^ab^	73.64 ^b^	75.84 ^b^	72.95 ^b^	1.88	0.29	0.02
GE								
d 23 to 27	74.46 ^a^	85.25 ^b^	85.54 ^b^	86.75 ^b^	65.63 ^c^	1.18	0.33	0.04
d 64 to 68	78.22	80.64	75.64	80.42	79.37	2.05	0.72	0.95

Note: Means within the same row with different superscript letters differ significantly (*p* < 0.05). Orthogonal contrasts: L = linear and Q = quadratic effect of yeast culture supplementation.

**Table 5 animals-12-01270-t005:** Effect of yeast culture supplementation in goose diets on serum parameters.

Parameter	Yeast Culture Supplementation (% of Diet)					SEM	*p*-Value	
	Control	0.5%	1.0%	2.0%	4.0%		L	Q
ALT (U/L)								
d 28	12.63 ^a^	20.36 ^b^	20.04 ^b^	23.27 ^b^	22.12 ^b^	0.68	0.22	0.15
d 70	14.26 ^a^	16.53 ^b^	15.17 ^ab^	29.05 ^c^	29.42 ^c^	0.42	0.05	0.17
AST (U/L)								
d 28	17.43 ^a^	13.23 ^b^	13.17 ^b^	14.75 ^b^	14.21 ^b^	0.68	0.65	0.68
d 70	22.20 ^a^	16.45 ^b^	26.27 ^c^	41.43 ^d^	41.02 ^d^	0.62	0.07	0.2
ALP (U/L)								
d 28	583.11 ^a^	985.63 ^b^	993.41 ^b^	658.85 ^c^	692.58 ^d^	3.46	0.69	0.83
d 70	786.25 ^a^	1061.63 ^b^	848.37 ^d^	853.74 ^c^	850.02 ^cd^	1.45	0.78	0.95
TC (mmol/L)								
d 28	6.40 ^a^	5.76 ^b^	7.06 ^c^	6.07 ^d^	6.27 ^e^	0.03	0.92	0.99
d 70	4.76 ^a^	5.11 ^b^	4.21 ^c^	4.21 ^c^	4.22 ^c^	0.01	0.24	0.41
TG (mmol/L)								
d 28	0.50 ^a^	0.55 ^b^	0.69 ^c^	0.55 ^b^	0.61 ^d^	0.01	0.58	0.77
d 70	0.59 ^a^	0.55 ^b^	0.78 ^c^	0.91 ^d^	0.93 ^d^	0.01	0.06	0.12
HDL (mmol/L)								
d 28	4.65 ^a^	4.09 ^b^	5.15 ^c^	4.26 ^b^	4.58 ^a^	0.08	0.97	1.0
d 70	3.24 ^a^	3.61 ^b^	2.91 ^c^	2.66 ^d^	2.64 ^d^	0.01	0.13	0.32
LDL (mmol/L)								
d 28	1.98 ^a^	1.70 ^b^	2.39 ^c^	1.91 ^d^	1.87 ^e^	0.01	0.82	0.9
d 70	1.61 ^a^	1.75 ^b^	1.42 ^c^	1.64 ^a^	1.61 ^a^	0.01	0.95	0.96

Note: Means within the same row with different superscript letters differ significantly (*p* < 0.05). Orthogonal contrasts: L = linear and Q = quadratic effect of yeast culture supplementation.

**Table 6 animals-12-01270-t006:** Effect of yeast culture supplementation in goose diets on immune factors.

Item	Yeast Culture Supplementation (% of Diet)					SEM	*p*-Value	
	Control	0.5%	1.0%	2.0%	4.0%		L	Q
Lysozyme (μg/mL)								
d 28	0.79 ^b^	0.65 ^a^	0.78 ^b^	1.01 ^c^	0.91 ^bc^	0.06	0.25	0.47
d 70	0.40 ^a^	0.73 ^b^	0.79 ^b^	1.09 ^c^	0.48 ^a^	0.04	0.99	0.04
IL-2 (ng/mL)								
d 28	46.12 ^a^	50.65 ^b^	64.93 ^c^	68.81 ^d^	83.81 ^e^	0.28	0.01	0.04
d 70	17.62 ^a^	20.02 ^b^	18.97 ^ab^	19.72 ^b^	17.50 ^a^	0.63	0.61	0.32
IgA (g/L)								
d 28	0.32	0.33	0.33	0.33	0.33	0.01	0.36	0.35
d 70	0.33	0.34	0.33	0.34	0.34	0.004	0.31	0.63
IgG (g/L)								
d 28	2.34	2.54	2.48	2.49	2.39	0.15	0.83	0.46
d 70	2.44 ^a^	1.65 ^b^	2.23 ^a^	2.50 ^a^	2.31 ^a^	0.11	0.65	0.92
IgM (g/L)								
d 28	0.15 ^a^	0.30 ^b^	0.18 ^a^	0.28 ^b^	0.30 ^b^	0.02	0.28	0.59
d 70	0.20 ^a^	0.33 ^b^	0.15 ^c^	0.32 ^b^	0.31 ^b^	0.02	0.45	0.8
Complement C3 (g/L)								
d 28	0.13 ^a^	0.19 ^b^	0.15 ^ab^	0.15 ^ab^	0.13 ^a^	0.01	0.52	0.72
d 70	0.20 ^a^	0.18 ^a^	0.24 ^b^	0.18 ^a^	0.27 ^b^	0.01	0.22	0.45
Complement C4 (g/L)								
d 28	0.08 ^a^	0.06 ^ab^	0.08 ^a^	0.04 ^b^	0.06 ^ab^	0.01	0.42	0.51
d 70	0.07 ^a^	0.08 ^a^	0.07 ^a^	0.07 ^a^	0.04 ^b^	0.01	0.05	0.06

Note: Means within the same row with different superscript letters differ significantly (*p* < 0.05). Orthogonal contrasts: L = linear and Q = quadratic effect of yeast culture supplementation.

## Data Availability

The datasets used and/or analyzed during the current study are available from the corresponding author upon reasonable request.

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
