# Peer review of "Effects of Yeast Culture Supplementation on Growth Performance, Nutrient Digestibility, Blood Metabolites, and Immune Response in Geese"

_animals, 2022, doi:10.3390/ani12101270_

Round 1

Reviewer 1 Report

  1. The Abstract should contain information regarding the levels of YC supplemented experimentally. The 2% inclusion is mentioned several times, but the reader does not have information regarding the (lack of) response to the other levels. Reading the Abstract, the reader is induced to believe that 2% of YC was the maximum level.
  2. Line 90 - "completely randomly " would be "completetely randomized design"?
  3. Table 1. Choline chloride 0.05 0.12. Please, check and/or justify the supplementation of choline. The usual is to redece the inclusion in grower feed compared to starter feed.
  4. Table 1. These levels are not correct: The premix provide the following per kilogram of diet: VA 2,000 IU, VD3 1,000 IU, VE 3,000 mg,
    VK3 200 mg, VB1 100 mg, VB2 1200 mg, VB6 200 mg, VB12 2.5 mg, nicotinic acid 600 mg, pantothenic acid 1,800 mg, folic acid 200 mg, biotin 20 mg, Fe 6 g, Cu 0.2 g, Mn 15 g, Zn 8 g, I 10 mg, Se 30 mg. The vitamin A and vitamin D are correct, but not the others. Are they the concentration in premix? Check and correct.
  5. The paragraphs from line 110 to line 127 are not well written and need to be improved in terminology, language.
  6. One important information is missing: YC was included on teh basal diet? Or there was a different technique of supplementation?
  7. Results section: Table 3. I strongly recomment that the data for BW (in grams) at 28 and 70 d have NO decimal point. The data for ADFI in tha last two periods should have only one decimal point. The p-values should be written in the conventional format (p<0.01 or p<0.001) - exact p value is not very informative.
  8. Reading the results (lines 163-174) reinforces the need of providing information of significant diferences obtained with 0.5, 1.0 or 4.0% YC for different variables for the reader of the Abstract.
  9. Lines 169-170:  "As the dietary YC level increased, improvement in growth performance efficiency first increased then decreased" is very confusing. Must be rewritten.
  10. Nutrient digestibility lines 176-185: the variations with levels were described - why regression analysis was not performed if the interest was to describe these variations?
  11. Lines 202-203;"As the dietary YC increased, IL-2 was increased on day 28." Again: regression analysis gives a better description of the trend.

Reviewer 2 Report

Effects of yeast culture supplementation on growth performance, nutrient digestibility, blood metabolites, and immunomodulation in geese

I read this manuscript for possible publication in Animals. Presented manuscript concerns on important and interesting subject which is interesting for breeders.

In general, the organization and the structure of the article are in agreement with the journal's instructions for authors.

But, I would suggest a modified the manuscript with minor corrections. I have indicated  my detailed  remarks and suggestions below:

1) Title: replace this word “immunomodulation” with another

2) In the Abstract: lines 28-29: the sentence should be as follows: “Calcium, phosphorus, crude protein and gross energy digestibility were increased.”

Line 38 – taken together change to in conclusion

3) Keywords: it would be good if the keywords were not the same as in the manuscript title and “geese” should be at the end

4) The introduction is well described, short, concise, and contains all the information you need about the study.

5) Materials and Methods:

lines 89-90: should be: At total, three hundred healthy Sichuan white goose (mixed sex) at 1-day-old age having an  average body weight (BW) of 95.57 ± 2.42 g were used and next  randomly divided  into five groups containing 0% (control), 0.5%, 1.0%, 2.0% and 4.0%  of a commercial  YC (Baihuibang, Beijing Enhalor Biotechnology Co., Ltd., Beijing, China) product in the diet.

Lines 99-102: should be: The basal diets with corn-soybean meal in recommendations of the NRC (1994) for starter (1-28 days) and for grower (29-70 days) periods were formulated  (Table 1). Geese were housed from April to June in pens (3.5 m × 3.0 m) and  rearing in a windowed poultry house.

Lines 103-104: should be: Geese were allowed access to feed (in pellet form) and water ad  libitum throughout the whole experimental period.

There is no information in the Materials and Methods regarding the consent of the ethics committee to this type of research.

The whole subheading 2.2. Sampling and Analyses should be modified with appropriate subheadings describing the analysis data.

Lines 110-111: should be: Every morning  the feed consumption before the feeding were recorded  and body weight (BW) was measured on day 1, 28 and 70 with empty stomach.

Lines: 115-116: should be: Determination of nutrient digestibility was by collecting ......

The entire paragraph on lines 118-126 should be re-described with better understanding.

The same with paragraph in lines 128-136.

Lines 141-142: should be: All the above tests  were carried out using commercial kits (Solarbio, Beijing, China).

Lines 143-144: should be rewritten.

Line 145: delete: In brief

Line 152: should be: were shown in Table 2.

Line 159: should be: data was expressed ....

6) Results

I would put Mortality at the end of the paragraph.

The entire paragraph describing growth performance should be slightly modified.

Table 4, 5 and 6 should be in separate brackets.

Whole paragraph about nutrient digestibility should be modified.

Line 190: instead of on the whole it should be in general.

The paragraph of immune factors should be also modified, that to be more understandable.

IFN-γ gene expression levels were significantly increased at 1.0% and 2.0% of what? And the same in whole paragraph.

Generally, the description of the results should be modified for better understanding.

7) Discussion

I wouldn't start the discussion with mortality, that wasn't the main focus of research. I would put the entire first paragraph at the end of the discussion.

Line 232: Here? What is this statement? Should be: In present study or current study....

8) Conclusions

Develop this paragraph.

9) References

59 references are a bit too much for an original article. I propose to delete the oldest literature.
